# Glioma–Immune Cell Crosstalk in Tumor Progression

**DOI:** 10.3390/cancers16020308

**Published:** 2024-01-11

**Authors:** Mahmoud Elguindy, Jacob S. Young, Isha Mondal, Rongze O. Lu, Winson S. Ho

**Affiliations:** Department of Neurological Surgery, University of California, San Francisco, CA 94143, USA

**Keywords:** glioma, immune, immune evasion, tumor microenvironment, immunotherapy

## Abstract

**Simple Summary:**

The environment surrounding primary glioma brain tumors plays a critical role in tumor development and progression. Immune cells are important constituents of this tumor microenvironment. Classically, immune activation is thought to halt tumor progression; however, recent evidence has shown that glioma cells can interact with immune cells to convert them into tumor-supporting cells. These reprogrammed immune cells allow glioma cells to evade the anti-tumor effects of the immune system and can even promote tumor growth. Understanding the mechanisms that result in this altered immune tumor microenvironment may pave the way to new therapies for brain tumors. This review aims to provide an updated view of the mechanisms underlying glioma–immune cell interactions and novel therapies being developed to combat glioma immune evasion.

**Abstract:**

Glioma progression is a complex process controlled by molecular factors that coordinate the crosstalk between tumor cells and components of the tumor microenvironment (TME). Among these, immune cells play a critical role in cancer survival and progression. The complex interplay between cancer cells and the immune TME influences the outcome of immunotherapy and other anti-cancer therapies. Here, we present an updated view of the pro- and anti-tumor activities of the main myeloid and lymphocyte cell populations in the glioma TME. We review the underlying mechanisms involved in crosstalk between cancer cells and immune cells that enable gliomas to evade the immune system and co-opt these cells for tumor growth. Lastly, we discuss the current and experimental therapeutic options being developed to revert the immunosuppressive activity of the glioma TME. Knowledge of the complex interplay that elapses between tumor and immune cells may help develop new combination treatments able to overcome tumor immune evasion mechanisms and enhance response to immunotherapies.

## 1. Introduction

Glioma development and progression occur in concert with continuous alterations in the surrounding tumor microenvironment (TME). The TME is a heterogeneous milieu of cells consisting of cancer cells and nonmalignant cells, including stromal cells, fibroblasts, endothelial cells, and immune cells integrated in a complex extracellular matrix [1]. Recent studies have focused on investigating the role of the TME in tumor development and how these cells interact with each other and with the malignant cellular component [1,2,3]. There is increasing evidence that glioma cells can functionally sculpt the TME through secretion of cytokines, chemokines, and other factors that ultimately reprogram components of the TME to promote tumor survival and progression [1,4,5,6].

Immune cells are important constituents of the TME and play key roles in surveillance against the development and advancement of glioma. However, growing evidence has shown that glioma cells can also co-opt innate immune cells (macrophages, neutrophils, dendritic cells (DCs), myeloid-derived suppressor cells, and natural killer (NK) cells) as well as adaptive immune cells (T and B cells) to promote tumor progression [1,6,7,8]. A plethora of mechanisms regulated by tumor-associated cells enable the immune escape of glioma cells and potentiate an immune-suppressive TME. Understanding these altered crosstalk interactions between glioma and immune cells may help develop new treatments to overcome tumor immune evasion.

In this review, we summarize the current knowledge on glioma crosstalk with various immune cells in the TME that contribute to the reprogramming of the TME and promote tumor progression and clinical strategies targeting these interactions.

## 2. Myeloid Cells

### 2.1. Tumor-Associated Macrophages and Microglia (TAMs)

Macrophages in the brain consist of both bone marrow-derived macrophages (BMDMs) and resident macrophages of the central nervous system (CNS), termed microglia. Studies have revealed that microglia originate from yolk sac progenitors during embryogenesis and have distinct transcriptional states depending on their topological distribution in the human brain [9,10,11]. While the exact differences in microglia and BMDMs remain to be elucidated, these macrophage types modulate immune responses through pathogen phagocytosis and antigen presentation and also function in wound healing and tissue repair. Simplistically, macrophages are dichotomized into M1 pro-inflammatory (i.e., anti-tumor) and M2 alternatively activated, anti-inflammatory (i.e., pro-tumor) phenotypes. However, more recent evidence has suggested that macrophages display a complex profile of both M1 and M2 polarization markers and likely exist in a continuum rather than in two distinct states [12]. In gliomas, tumor-associated macrophages (TAMs) are the most abundant population of TME immune cells, making up to 50% of total live cells in gliomas [13]. TAMs release a wide array of growth factors and cytokines in response to factors produced by cancer cells, playing a critical tumor-supportive role in the TME. This direct, pro-tumorigenic effect of TAMs is illustrated in mouse models of glioma where perturbation of microglia/macrophage function attenuates glioma proliferation [14,15,16]. Moreover, high macrophage infiltration in gliomas correlates with a negative prognosis [17], similar to other tumor types [18], further establishing their role in cancer progression. Recent studies have begun to try to distinguish between microglia and BMDMs in glioma formation and progression. Lineage-tracing experiments in mouse glioblastoma models have suggested that 85% of TAMs are BMDMs, which are predominantly localized in the perivascular areas of the tumor, while microglia are peri-tumoral [19]; however, other studies have suggested that the majority of TAMs are intrinsic microglia [20]. Interestingly, studies comparing the type of TAM (i.e., microglia vs. BMDM) in IDH-mutant and wild-type tumors have suggested that IDH-mutant TAMs were composed primarily of microglia, while IDH-wild-type tumors had more BMDM infiltration [12,21]. In astrocytomas, increased infiltration of TAMs derived from BMDM was associated with higher tumor grade and reduced survival [22]. More in-depth studies are needed to accurately distinguish microglia from BMDMs and further elucidate their distinct roles within specific types of gliomas. Given the current lack of clear evidence on the differences in roles between microglia and BMDMs in glioma, our connotation of TAMs in this review denotes macrophages without distinction between these two myeloid cell populations.

#### 2.1.1. Glioma Effect on TAMs

Gliomas help recruit TAMs to the TME by secreting various chemokines, including monocyte chemoattractant proteins (MCP-1/CCL2 and MCP-3) [23], C-X-C motif chemokine 12 (CXCL12) [24], colony-stimulating factor 1 (CSF-1) [25], lysyl oxidase (LOX) [26], and glial cell-derived neurotrophic factor (GDNF) [27] (Figure 1). Genomic alterations in gliomas can lead to upregulation of these chemokines and increased recruitment of TAMs. For example, neurofibromatosis-1 (NF1) deficiency in gliomas is associated with increased infiltration of TAMs [28]. Chen et al. also demonstrated that genomic alterations in the PTEN-phosphatidylinositol 3-kinase (PI3K) pathway, commonly mutated in glioblastoma (GBM) cells, lead to enhanced recruitment of TAMs. Mechanistically, this effect was mediated through the activation of Yes-associated protein 1 (YAP1), which transcriptionally upregulates LOX expression and secretion from tumor cells. Subsequently, LOX functions as a potent macrophage chemoattractant via activation of the β1 integrin-proline-rich tyrosine kinase 2 pathway (PYK2) in macrophages [26]. Additionally, amplification of epidermal growth factor receptor (EGFR) and its truncation mutant variant (EGFRvIII) induces MCP-1 expression and secretion, promoting TAM recruitment [29].

Gliomas can also reprogram TAMs in the TME toward an immunosuppressive, tumor-promoting phenotype. Glioma-derived CSF-1 not only acts as chemoattractant but also induces a shift of microglia and macrophages toward a pro-tumor phenotype, and blocking CSF-1 receptor (CSF-1R) inhibits alternative activation of TAMs and attenuates glioma progression [30]. However, gliomas can acquire resistance to CSF-1R inhibition mediated by changes in the TME that lead to increased secretion of interleukin (IL)-4, a potent inducer of pro-tumor, alternatively activated TAMs, from CD8+ T cells and other cell types [31]. Upregulation of IL-4/IL-4R signaling in TAMs results in their increased secretion of insulin-like growth factor (IGF-1), which, upon binding to IGF-1 receptor (IGF-1R) on tumor cells, upregulates the PI3K pathway, promoting tumor growth and malignancy [31]. Accordingly, halting this feedforward loop of TAM recruitment and alternative activation through a combination of CSF-1R and IGF-1R inhibition leads to a further reduction in tumor progression in mice [31]. In addition to CSF1 and IL-4-mediated TAM reprogramming, the chemokine CXCL16 released from tumor cells has also been shown to bind to CXCR6 on microglia cells and drive them toward a pro-tumor phenotype [32]. Another feedforward mechanism involves glioma-derived MCP-1/CCL2, which triggers the release of IL-6 from microglia, a ligand for signal and activator of transcription 3 (STAT3) pathway in gliomas [33]. STAT3 induces a variety of transcriptional factors that propagate tumorigenesis, including maintaining the cancer stem cell state and invasiveness and upregulation of immunosuppressive factors. In glioma stem cells (GSCs), STAT3 activation leads to increased secretion of macrophage inhibitory cytokine-1 (MIC-1), IL-10, IL-4, and transforming growth factor β (TGF-β), which inhibit the phagocytic activity of macrophages and induce them to become immunosuppressive [34,35,36,37]. In addition to reprogramming of TAMs, glioma cells can upregulate the antiphagocytic “don’t eat me” surface protein CD47, which binds to its receptor signal-regulatory protein alpha (SIRP⍺) on TAMs and inhibits their phagocytic activity [38,39]. Blocking this myeloid checkpoint CD47-SIRP⍺ axis using anti-CD47 antibody therapy was shown to limit tumor growth in both adult and pediatric gliomas [40,41] and also promoted M1 activation of TAMs [42].

#### 2.1.2. TAMs’ Effect on Glioma

A number of TAM-secreted factors, including stress-inducible protein 1 (STI1) [43], epidermal growth factor (EGF) [25], TGF-β [44,45], and matrix metallopeptidase-2 (MMP-2) [46], have been shown to enhance proliferation and invasiveness of glioma cells (Figure 1). The release of cytokines, including IL-12, IL-1B, CCL8, and IL-6 by TAMs, has also been shown to promote glioma stem cell renewal [13,47,48]. In addition, Shi et al. identified pleiotrophin as a TAM-secreted factor that activates AKT signaling in glioma cells and promotes tumor growth and stem cell renewal [49]. Glioma-induced polarization of macrophages also promotes macrophage secretion of osteopontin (SPP1) [26] and the expression of vascular endothelial growth factor (VEGF) in a receptor for advanced glycation endproduct (RAGE)-dependent manner [50], both of which promote glioma cell survival and angiogenesis.

TAMs also play important roles in creating an immunosuppressive environment for glioma immune evasion. TAM expression of chemokines such as CCL2, 5, 20, and 22 increase recruitment of regulatory T cells (Tregs), which subsequently inhibit the activity of effector T cells, natural killer (NK) cells, and antigen-presenting cells (APCs) [51,52,53]. Microglia also express and secrete Fas-ligand (FasL), which binds to Fas receptors and induces the apoptosis of invading T cells in gliomas [54,55]. Additionally, recently described overexpression of indoleamine 2,3-dioxgynase 1 and tryptophan 2,3-dioxygenase 2 (IDO1/TDO2) in GBM promotes immunosuppression through the production of tryptophan metabolite L-Kynurenine (Kyn) [56]. Kyn can interact with aryl hydrocarbon receptor (AHR) on TAMs and other immune cells to repress pro-inflammatory cytokine release by TAMs and drive CD39 expression in TAMs, which contributes to T cell dysfunction while promoting Treg generation in GBM [56,57].

Recent spatial transcriptomic analysis has also highlighted distinct regions within glioma tumors that are enriched in myeloid and lymphoid cells and display mesenchymal-like transcriptional signatures [58], suggesting that these immune cells also genetically alter tumor cells. Indeed, evidence from Hara et al. has shown that macrophage-derived oncostatin can interact with receptors on glioblastoma cells and induce a mesenchymal-like transcriptional program through STAT3 activation [59]. Additionally, studies by Gangoso et al. have demonstrated that macrophage infiltration and interferon (IFN)-γ signaling induce a transcriptional and epigenetic reconfiguration in GSCs toward a mesenchymal-like profile, which leads to further recruitment of TAMs and immune evasion [60]. Correspondingly, this acquired mesenchymal-like state has been shown to be associated with poor prognosis and tumor recurrence [61]. Taken together, these studies highlight the bidirectional cellular interdependence between glioma cells and TAMs that help establish an immunosuppressive TME and promote tumorigenesis.

### 2.2. Neutrophils

Neutrophils account for up to 70% of circulating leukocytes and are classically the first responders to acute inflammation [62]. Upon tissue damage or infection, epithelial cells secrete neutrophil-homing chemokines, inducing them to extravasate from circulation and enter the damaged tissue where they perform various functions, including phagocytosis of invading microorganisms, secretion of inflammatory cytokines (e.g., tumor necrosis factor alpha (TNF⍺), IL-4, IL-8), and formation of neutrophil extracellular traps (NETs) [62]. Current cancer literature suggests an important role for neutrophils within the TME [63]. Studies on various cancer types have suggested that neutrophil expansion, both in the TME and systemically, is associated with poor prognosis [63]. In tumors, neutrophils are dichotomized into N1 (tumor-suppressive or high-density neutrophils (HDNs)) and N2 (tumor-promoting or low-density neutrophils (LDNs)) phenotypes. In many cancer types, LDNs, which exhibit a more immature phenotype, predominate in the circulation and may contribute to cancer progression and metastasis [64].

In the context of glioma, little mechanistic work has been performed on neutrophils or other granulocytes until recently. Early work showing that higher-grade gliomas have higher circulating and tumor-infiltrating neutrophil levels has suggested that neutrophils play an important role in glioma TME, and glioma-derived factors may affect these cell types [65]. Further assessment of the relationship between neutrophils and prognosis has demonstrated that an increased proportion of neutrophils to lymphocytes in peripheral blood is associated with poor prognosis in glioma patients [66]. Moreover, neutrophil-specific DNA methylation associated with dexamethasone treatment (NDMI score) was recently shown to be an accurate marker for patient survival, inversely correlating with prognosis and positively correlating with an immunosuppressive TME [67]. Gliomas can induce peripheral neutrophilia through the secretion of granulocyte-colony stimulating factor (G-CSF) and granulocyte-macrophage colony-stimulating factor (GM-CSF) [68]. Recent work by Lad et al. has also demonstrated that GM-CSF in the glioblastoma secretome enhances the longevity of peripheral blood neutrophils [69]. In addition to enhancing granulopoiesis, gliomas can recruit neutrophils to the TME through expression of high levels of IL-8, under stimulation of IL-1, TNF⍺, and high-mobility group box 1 (HMGB1) derived from NETs [70,71]. Additionally, in in vitro co-cultures, glioma-derived IL-6 and IL-8 were shown to extend neutrophil survival, suggesting glioma–neutrophil interactions [72]. Further studies have identified RAGE expressed on glioma tissues as a binding receptor for neutrophils, which subsequently activates NF-κB signaling and further promotes neutrophil infiltration [71].

With respect to the effect of neutrophils on gliomas, work by Liang et al. has shown neutrophils can promote the mesenchymal transformation of gliomas via induction of S100A4 expression within glioma cells [73]. In contrast, other reports have also shown that neutrophils recruited during the early stages of glioma development exert an anti-tumor effect in mice through increased production of reactive oxygen species (ROS) [74]. However, this anti-tumor effect was lost with tumor progression. Interestingly, Lad et al. recently showed that a subset of TANs are recruited from skull marrow and uniquely differentiate into APC-like cells that activate T-cells and suppress tumor growth [69]. A more detailed understanding of neutrophils and other granulocytes and signals that pivot neutrophils to become immunosuppressive may hold promise for a better understanding of the reprogramming system of the TME and new potential immunotherapy targets.

### 2.3. Dendritic Cells (DCs)

Dendritic cells (DCs) are specialized antigen-presenting cells (APCs) that play key roles in the initiation of anti-tumor immune responses by linking innate and adaptive immunity [75]. DCs are not typically found in normal brain parenchyma but are present in vascular-rich compartments such as the choroid plexus and meninges [76,77]. In response to a variety of stimuli, including exposure to pathogens, nucleic acids, and type I interferons, DCs undergo activation and maturation, during which they acquire potent T-cell stimulatory ability [78]. While the specific role of DCs in the setting of gliomas is still being elucidated, current studies suggest an interplay between DCs, tumor cells, and TME. Studies on primary brain tumors and metastatic brain tumors have shown that DCs can recognize and traffic tumor antigens to tumor-draining deep cervical lymph nodes to elicit T cell-mediated responses [79,80,81]. They also can produce chemokines such as CCL9 and CCL10 that can recruit cytotoxic T lymphocytes (CTLs) into the tumor [81,82]. Furthermore, upon stimulation by T cell-produced IFN-γ, DCs can subsequently boost the anti-tumor activity of T and NK cells through the production of cytokines such as IL-12 [83]. It has been noted that the lymphatic migration of DCs to and from tumors and to draining cervical lymph nodes may be a critical aspect underlying response to immunotherapies, such as immune checkpoint inhibitors. Song et al. demonstrated that enhancement of lymphatic vasculature in the brain through VEGF-C expression synergizes with immune checkpoint blockade to elicit a robust adaptive immune response and improve survival outcomes in mouse glioma models [84]. Moreover, studies from Hu et al. have shown that VEGF-C-augmentation of lymphatics results in increased DC migration from the CNS to cervical lymph nodes in a CCL21/CCR7-dependent manner, and this DC migration is critical for the survival benefit seen with combined VEGF-C and immune checkpoint blockade therapy [85]. These findings highlight the essential role the lymphatic vasculature and DCs play in mediating anti-tumor responses.

Studies in other tumor types have identified various molecules found in the TME and expressed by tumor cells that inhibit DC activation and drive DCs toward an immunosuppressive, regulatory phenotype, including vascular endothelial growth factor (VEGF), prostaglandin E2 (PGE2), IL-10, and CSF-1 [86,87,88]. These regulatory DCs can then promote Treg activation and downregulate the recruitment and activity of CTLs by secreting cytokines such as IL-10 and TGF-β [89,90]. Nrf2, a redox-sensitive transcription factor expressed in DCs and whose overexpression positively correlates with glioma progression [91], was recently shown to suppress DC maturation and consequently result in decreased T cell activation [92]. Moreover, recent work on DC differentiation in IDH wild-type and mutant gliomas has also shown that wild-type tumors exhibit higher infiltration of DCs, while production of R-2-hydroxyglutarate (R-2-HG) in IDH-mutant gliomas impairs dendritic cell differentiation and maturation and suppresses major histocompatibility complex (MHC I/II) presentation, resulting in reduced T-cell activation [93]. Further investigation is needed into the specific roles of DC in gliomas and the complex interplay between DCs, tumor cells, and other immune cells.

### 2.4. Myeloid-Derived Suppressor Cells (MDSCs)

Myeloid-derived suppressor cells (MDSCs) are a heterogeneous population of myeloid-origin TME cells comprising progenitor and immature macrophages, granulocytes, and DCs. In patients with gliomas, peripheral blood MDSCs are elevated compared to non-tumor samples [94]. Additionally, the intratumoral density of MDSCs increases with glioma progression and correlates with patient survival [95,96]. MDSCs have been shown to be powerful inhibitors of anti-tumor immune responses through various mechanisms discussed below.

A variety of inflammatory mediators can induce MDSC expansion and recruitment to the TME, including tumor-derived IL-6, IL-8, IL-10, CSF-1, CCL2, CXCL2, PGE2, and TGF-β [97]. Gliomas can also increase intracellular levels of S100A8/9 (calprotectin) in MDSCs, an important marker of MDSC development and activation [98]. In addition, hypoxic conditions in the TME can alter the metabolism of MDSCs toward fatty acid oxidation, prompting upregulation of arginase I (Arg1) and nitric oxidase in MDSCs, which aid in their ability to suppress the immune system [99].

MDSCs present in the TME contribute to immunosuppression through suppression of T cells, NK cells, macrophages, and DCs functions and induction of Tregs [100,101,102]. While the exact roles MDSCs play in gliomas have yet to be fully elucidated, work in other tumor types has shown that MDSCs can induce oxidative stress by producing nitric oxide (NO), which can have several downstream effects. NO can upregulate Notch signaling and IL-6 signaling in tumor cells, leading to prolonged STAT3 activation and promoting cancer cell stemness [103]. In NK cells, MDSC-released NO reduces the cytotoxicity of NK cells and inhibits their ability to secrete IFN-γ and TNF⍺ [104]. Oxidative stress also potently blocks T-cell proliferation and activation by nitration/nitrosylation of chemokines and T-cell receptors [105]. TGF-β release by MDSCs has also been shown to induce Treg differentiation, NK cell anergy, and M2 macrophage polarization [106,107]. MDSCs are known to deplete essential metabolites in the TME, including arginine depletion promoted by Arg1 activity, which alters T cell receptor formation and induces T cell proliferation arrest [108,109]. Further work is needed to investigate the plethora of mechanisms through which MDSCs modulate the immune system in gliomas.

## 3. Lymphocytes

T cells, including CD8+ cytotoxic T lymphocytes (CTLs), regulatory CD4+ (Treg), and conventional CD4+ T cells, constitute 1–5% of the total glioma cellular content [110,111]. CTLs are considered the most potent anti-tumor cells, and many emerging immunotherapies have focused on enhancing CTL activity [112,113]. Upon priming and activation by antigens bound to major histocompatibility locus (MHC) I proteins expressed on APCs, CD8+ T cells differentiate into CTLs. CTLs can then release perforin- and granzyme-containing granules, resulting in an efficient anti-tumoral attack with direct destruction of target cells. CD4+ T helper 1 (Th-1) cells also mediate an anti-tumor response through the secretion of proinflammatory cytokines such as IL-2, TNF⍺, and IFN-γ, which promotes not only T-cell priming and CTL cytotoxicity but also the anti-tumoral activity of macrophages and NK cells [114,115]. Although debated in gliomas, the presence of tumor-infiltrating CD8+ T cells and Th-1 cytokines in tumors correlates with a favorable prognosis in terms of overall survival and disease-free survival in many malignancies [116].

A hallmark of tumor cells is the evasion of this lymphocyte-induced immune attack using two main strategies: avoiding immune recognition and instigating an immunosuppressive TME, including impairing the effector functions of anti-tumor CD4+ and CD8+ T cells and promoting immunosuppressive Treg and Th17 cells [117]. Below, we discuss the tumor–lymphocyte interactions that are involved in glioma immune evasion.

### 3.1. T cells

#### 3.1.1. Reprogramming of T Cells

Gliomas can reprogram T cells into a dysfunctional state through multiple mechanisms (Figure 2). First, gliomas and tumor-associated cells, including lymphocytes, can express immune checkpoint molecules such as programmed death receptor Ligand-1 (PD-L1) [118,119,120], which binds to Programmed Death Receptor-1 (PD-1) on T-cells, blocking T-cell differentiation and activation. Notably, the expression of PD-L1 in gliomas is heterogeneous and relatively low, and evidence suggests that it is primarily expressed in lymphocytes [120] and myeloid cells in gliomas [121]. PD-L1 expression was noted to be lower in IDH-mutant, low-grade gliomas (LGG) [119], which also had reduced expression of T-cell lymphocyte-associated genes compared to IDH-wild-type, high-grade gliomas (HGG) [122]. Further investigation is needed to elucidate the mechanisms contributing to these differences in LGG versus HGG checkpoint molecule expression. Second, the glioma TME can recruit and promote Tregs to secrete immunosuppressive cytokines, as discussed below, which further suppress effector T cell function. Third, various factors secreted in the TME, such as NO by MDSCs and IL-10 and TGF-β from Tregs, induce T cell tolerance, whereby T cells do not respond to antigen stimulation due to the presence of immunosuppressive cytokines or signals [105,123]. Ultimately, T cell exhaustion may occur from persistent antigen stimulation of naïve T cells, which is characterized by reduced expression of effector molecules with increased expression of checkpoint molecules, including PD-1 and cytotoxic T-lymphocyte-associated protein 4 (CTLA-4) [124,125].

#### 3.1.2. Evasion of Lymphocyte Surveillance

Gliomas, like many other tumors [126], can evade immune surveillance by downregulation of MHC-I expression [127]. MHC-I downregulation reduces the number of tumor-infiltrating lymphocytes and correlates with a worse prognosis [126]. In general, tumors may reduce MHC-I antigen expression through genomic defects perturbing heavy a-chain (i.e., HLA-encoded) or light chain (i.e., B2-microglobulin, B2M) gene expression or transcriptional (e.g., downregulation of type I and II IFN pathways, loss of transcriptional regulators, DNA hypermethylation or histone deacetylases) and post-transcriptional (e.g., microRNAs) silencing [128]. While the exact mechanisms underlying MHC-I downregulation in gliomas remain to be elucidated, a recent paper by Mondal et al. identified an intimate relationship between type I IFN signaling and MHC-I expression in glioma [129]. Specifically, they found that inhibition of tumor-expressed protein phosphatase-2A (PP2A), a serine-threonine phosphatase involved in DNA damage response and inhibition of which had previously been shown to enhance tumor immunity [130,131], leads to the accumulation of cytosolic double-stranded DNA and consequent induction of the cGAS-STING pathway. STING activation then induces phosphorylation of interferon regulatory factor 3 (IRF3) to promote type I IFN production and downstream JAK/STAT1 activity, leading to upregulation of MHC-I expression.

#### 3.1.3. Recruitment of Tregs

Cancer cells and the TME simultaneously suppress anti-tumor T cell function through the recruitment of Tregs. Patients with malignant gliomas have an increased proportion of immunosuppressive Tregs and reduced CD4+ T cell number [132]. Glioma cells can recruit Tregs to the TME through the secretion of CCL22 [133,134] and other factors [135]. Indoleamine 2,3-dioxygnase 1 (IDO1) expression in gliomas has been suggested to be another important Treg recruiter, potentially via chemokine induction, and IDO upregulation is associated with a decrease in survival in glioma patients [136].

Tregs are known to exert immunosuppression through effects on numerous TME cell types [137] via various mechanisms, including consumption of IL-2, thereby suppressing T cell activation [138]; production of immunosuppressive cytokines such as IL-10, IL-35, and TGF-β [133,139,140]; conversion of ATP to adenosine, which hinders T cell activation [141]; CTLA-4-mediated suppression of APC function [142]; and secretion of granzyme or perforin to destroy effector cells [143]. While the induction of Treg activity is important for immune evasion by gliomas and correlates with glioma progression [144], the contribution of each of these mechanisms in gliomas has yet to be elucidated.

### 3.2. Natural Killer (NK) Cells

NK cells are circulatory, innate lymphoid cells recognized for their cytotoxic and cell-mediated killing functions. NK cells are important regulators of tumor surveillance, as illustrated by a correlation between low NK cell activity and increased cancer risk [145]. NKs employ death receptor-mediated apoptosis and perforin/granzyme-mediated cytotoxicity to target tumor cells and limit primary tumor growth [51].

Limited studies have examined the direct effect of glioma on NK function. Nonetheless, extrapolation of evidence from studies on other tumor types suggests that several factors in the glioma TME limit NK cell activity. TGF-β has been shown to negatively regulate NK cytotoxicity and proliferation [146]. TGF-β was also shown to downregulate the expression of NKG2D activating receptors on NK cells obtained from peripheral blood in GBM patients [147]. Additionally, PGE2 secreted by tumor cells can signal through a G-protein coupled receptor EP4 to inhibit IFN-γ production and anti-tumor activity of NK cells in mice [148]. IDO-production of Kyn by gliomas has also been shown to downregulate the expression of activating NKG2D and NKp46 receptors on NK cells [56,57,149]. Adenosine [150] and lactate [151] metabolites present in the TME have also been shown to reduce NK cytotoxicity and maturation.

## 4. Immunotherapies

### 4.1. Cytokine and Molecular Therapies

Several cytokines fused to a specific tumor-associated epitope or delivered via oncolytic viruses have been investigated. IL2, IL12, and TNF⍺ were studied by Weiss et al., which demonstrated, in mice, that the delivery of each of these cytokines independently promoted CD4+ and CD8+ infiltration and reduced tumor volume, with the administration of IL-12 showing the highest cure rate [152]. Administration of TNF⍺ to human GBM patients increased tumor necrosis and tumor-infiltrating CD4+ and CD8+ T cells. However, the use of TNF⍺ therapy is limited due to systemic toxicities associated with the high levels needed to achieve efficacy [153]. Other cytokine/chemokine inhibitors have also been trialed with limited efficacy, such as a CSF1R inhibitor, which showed no improvement in progression-free or overall survival in the PLX3397 trial [154]. CCR2 inhibition targeting MDSC infiltration in gliomas did not show efficacy alone, but in combination with PD-1 blockade, was able to extend survival in murine glioma models [155]. A recent study utilizing recombinant IL-7, a cytokine that is required for T cell development, in GBM mouse models showed IL-7 treatment could expand CD8+ T cells in peripheral blood and the TME and improve overall survival in mice [156]. Other promising avenues of regulating cytokine levels in the TME include modulating intracellular pathways involved in immune suppression. For example, studies have shown that PP2A inhibition or activation of the STING pathway increases type I IFN signaling in TAMs, leading to increased activation of macrophages and CD8+ T cells and a subsequent reduction in tumor volume in mice [129,157].

### 4.2. Immune Checkpoint Inhibitors

Despite the notable success immune checkpoint inhibitors have shown in many cancer types, the same efficacy has not been observed in gliomas/glioblastomas. Unfortunately, three phase III trials assessing nivolumab, an anti-PD-1 therapy, alone or in combination with temozolomide or radiotherapy, failed to demonstrate a survival advantage in patients with GBM [158,159,160]. In addition, in a phase I trial, anti-PD-1 therapy in combination with anti-CTLA-4 therapy did not improve overall survival [161]. Nonetheless, there are some patients that show durable radiographic responses and prolonged survival following PD-1 blockade [162], suggesting that certain factors, such as PD-L1/PD-1 expression and patient-specific glioma-TME interactions, influence the efficacy of these treatments [163,164]. Indeed, recent studies looking into molecular and genetic factors associated with immune checkpoint inhibitor sensitivity identified patients with defects in replication stress response, a pathway important for maintaining genome stability and integrity during DNA replication, were highly sensitive to immune checkpoint blockade [165]. Moreover, results from Cloughesy et al. and Schapler et al. demonstrating that neoadjuvant PD-1 blockade can enhance the anti-tumor immune response [162,166] and improve overall and progression-free survival in patients with recurrent GBM [162], suggest that the timing of immune checkpoint blockade may also be an important factor affecting efficacy. Considering that current standard therapies for glioma patients are also immunomodulatory—such as temozolomide, which induces lymphopenia, which was shown to enhance antitumor activity associated with immunotherapies [167], and steroids, which are known to be immunosuppressive—further work is needed to investigate the optimal timing and order of immunotherapies given in these combination settings to achieve synergistic immune responses. Combination therapies with other immune checkpoint inhibitors, such as anti-CD47, which has shown promising results in adult and pediatric gliomas [40,41], may also be of benefit. In addition, further characterization of the molecular and immune landscape of immune checkpoint inhibitor-sensitive gliomas may allow for a more tailored and efficacious approach for utilizing these therapies as well as yield new targets that may be adapted to increase sensitivity to immune checkpoint blockade.

### 4.3. CAR T-Cell Therapy

CAR T therapy has emerged as a promising form of immunotherapy for various cancers. CAR T therapy involves engineering a patient’s own T cells to express a receptor specifically targeting antigens on tumor cells. Several trials using CAR T therapy in GBM have been completed or are currently active [168]. Early studies in mouse glioma models showed encouraging results for EGFRvIII-targeting CAR T cells, which led to cures in all mice with a sustainable response [169]. Subsequent studies on human patients demonstrated that while this therapy was safe and initially led to an anti-tumor response, disease recurrence, target-antigen loss, and escape were observed [170,171]. More recent work has focused on refining these CAR T cells to target multiple antigens, such as synNotch-CAR T targeting EGFRvIII and a brain-specific glycoprotein [172], to improve the specificity and persistence of these cells. Other studies have examined HER2 [173] and IL-13Ra2 [174], both highly expressed in GBM tissues, targeting CAR T cells in glioblastoma with promising initial results. For diffuse midline gliomas, the identification of a targetable, glioma-specific H3.3K27 mutation epitope [175] and the newly developed CAR-T targeting GD2 [176], a disialoganglioside highly expressed in H3K27M-mutated gliomas, hold significant promise for the use CAR T-based immunotherapy for patients with these tumors. More studies are also underway to develop modified CAR T cells with enhanced cytotoxicity and more durable responses [177,178]. Additional CAR T therapies for GBM and their potential limitations have been recently and comprehensively reviewed by Luksik et al. [168].

### 4.4. Oncolytic Viruses and Vaccines

Oncolytic viruses targeting tumor cells not only result in oncolysis of cancer cells but also augment host immune responses and promote proinflammatory pathways in the TME. Several oncolytic viruses have been and are currently being tested in gliomas [179]. DNX-2401, an adenovirus designed to be tumor-selective, has shown promising results in a phase I clinical trial on recurrent malignant glioma, with 20% of patients surviving greater than 3 years post-treatment and 12% showing at least a 95% reduction in tumor volume [180]. DNX-2401 was also recently tested in 12 pediatric diffuse intrinsic pontine glioma patients and was able to elicit immune-mediated anti-glioma response [181]. Trials examining engineered herpes simplex virus type 1 (HSV-1) oncolytic viruses have also reported encouraging results. Tumor-targeting HSV-1 G207 was shown to induce a strong immune response, with an increased number of tumor-infiltrating lymphocytes and probable improved survival in pediatric patients with high-grade gliomas [182]. Most recently, seminal work by Ling et al. has also demonstrated an HSV-1 oncolytic virus, CAN-3110, improved survival in recurrent GBM patients with HSV-1 seropositivity and dramatically increased tumor-infiltrating lymphocytes [183].

Vaccine-based therapies can elicit an anti-tumor immune response by introducing T cells to immunogenic tumor-specific or overexpressed antigens. Rindopepimut is an EGFRvIII-targeting peptide vaccine that was studied in a phase III clinical trial for newly diagnosed GBM patients who had undergone maximal resection and radiation therapy with temozolomide [184]. The trial showed that Rindopepimut alone did not improve overall survival and suggested that combination therapy with other immunotherapies may be of benefit. SurVaxM is another vaccine targeting a surviving peptide, an anti-apoptotic protein highly expressed in GBM cells. SurVaxM therapy was associated with higher median overall survival (86.6 weeks), and an ongoing clinical trial is evaluating this treatment in combination with pembrolizumab for recurrent GBM [185]. Other vaccine trials, including dendritic cell vaccines where DCs are generated to present antigens targeting GBM cells (e.g., HER2, IL13Ra2), have shown mixed results and are discussed in a comprehensive review by Sener et al. [186]

## 5. Conclusions and Perspectives

Cancer is no longer simply regarded as a genetic disease that leads to the formation of tumorigenic cells. Instead, we have come to appreciate the “ecological” aspect of this disease, where tumor development is intimately linked to reciprocal interactions with non-cancerous cells in their surrounding microenvironment [187]. Spatiotemporal interactions between these cells lead to adaptations and evolution of cancer as it progresses from initiation to immune evasion and invasion. The studies discussed above illustrate how gliomas sculpt an immune suppressive microenvironment that not only enables immune evasion but also promotes tumor growth and invasiveness. The infiltration of multiple immune cell types, including TAMs and Tregs, and their effect on neighboring cells through direct interactions and secreted cytokines, highlights the multidirectional, cellular interdependence the heterogenous TME exhibits. This complexity and multitude of immunosuppressive mechanisms within the TME may explain the limited efficacy of current immunotherapies.

Patient selection remains an important aspect to consider in the realm of immunotherapies. In this regard, it is important to note that the reciprocal interactions between glioma cells and the TME are continuously evolving from tumor initiation to progression and recurrence [188,189]. While the specific mechanisms underlying the evolution of glioma tumors remain to be solved, spatial transcriptomic and proteomic evidence has shown that glioma cells are dynamically reorganized into distinct regions that display unique molecular profiles or lineage states (e.g., proneural, mesenchymal, and classical) and are constantly adapting to evolving stresses from the surrounding TME [58,190]. Thus, the characterization of gliomas in patients cannot be a singular event and requires multiple regions and time points to be analyzed. With regard to surgery, this implicates the need for multiple tissue samples and serial biopsies and resections when treating patients with recurrent gliomas to best characterize their tumor-specific TME alterations and adapt immunotherapies based on this. In addition, recent studies have begun to examine changes in the immune TME associated with standard-of-care GBM therapies temozolomide and radiation [191]. Understanding how these treatments affect the TME and how gliomas may acquire resistance to these therapies through immune cell adaptations may help develop more efficacious combination treatments.

While significant progress has been made in characterizing the components of the TME, the majority of current studies, as highlighted above, have focused on the TAM population of immune cells, given their abundance in the TME. Further in-depth mechanistic studies on the role of other immune cell types in glioma biology are needed. Many fundamental aspects of immune cell–glioma interactions remain incompletely understood, such as the molecular changes involved in macrophage and DC reprogramming and NK cell suppression, the temporal alterations occurring in T cells over the course of glioma progression that lead to T cell exhaustion, the relationship between glioma mutations and immune cell recruitment and function, and, perhaps most importantly, how we may utilize genomic and transcriptomic analyses as approaches for precision medicine to develop more tailored and effective immune-based therapies. Defining these molecular underpinnings of glioma-TME crosstalk may lead to a new “magic bullet” therapeutic approach that interrupts the glioma ecosystem.

## Figures and Tables

**Figure 1 cancers-16-00308-f001:**
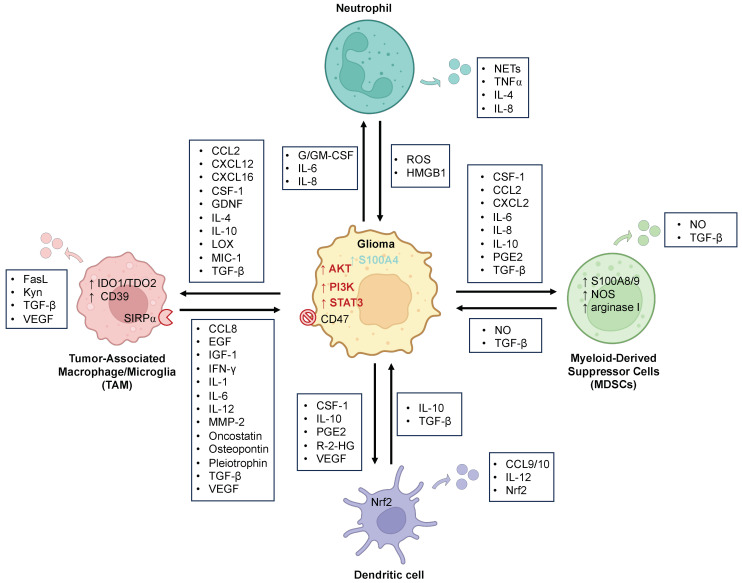
The bidirectional cellular interactions between myeloid and glioma cells. Myeloid and glioma cells exhibit numerous crosstalk interactions. Gliomas secrete various factors that help recruit and reprogram various myeloid immune cells. In turn, these reprogrammed myeloid cells can also secrete factors that promote tumor growth, transcriptionally rewire glioma cells towards a mesenchymal-like profile, and further recruit and reconfigure other immune cells in the TME, allowing tumor cells to evade and suppress the immune system. See text for abbreviations. Up and down arrows indicate upregulation and downregulation of the associated molecules or signaling pathways, respectively. Font color of signaling pathways upregulated in the schematized glioma cell corresponds to the associated immune cell color. Created with BioRender.com.

**Figure 2 cancers-16-00308-f002:**
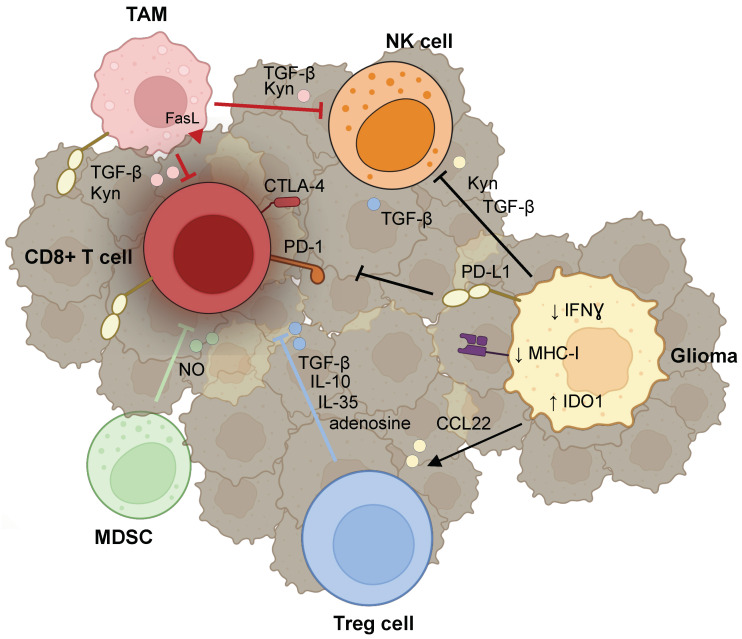
Immunosuppressive tumor microenvironment interactions limiting anti-tumor lymphocyte activity. Gliomas and other tumor-associated cells express various factors, including immune checkpoint proteins and cytokines, that result in dysfunctional, exhausted CD8+ T cells. In addition, gliomas sculpt the TME to promote recruitment of Treg cells, pro-tumor TAMs, and MDSCs, which further halt cytotoxic immune activation and promote tumor immune evasion. Up and down arrows illustrate upregulation and downregulation of the indicated molecules, respectively. Created with BioRender.com.

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
