# Peer review of "Glioma–Immune Cell Crosstalk in Tumor Progression"

_cancers, 2024, doi:10.3390/cancers16020308_

Round 1

Reviewer 1 Report

Comments and Suggestions for Authors

Major Comments

The authors prepared an interesting review article, however, the first major concern is that the review is both too broad and in some places too narrow to be a review on a specific topic. Would advise the authors to pick a focus of interest to narrow in on to provide a more comprehensive and appropriate review.

Of note, there are multiple review articles on TAMs and glioma/glioblastoma. There are less review articles focused on other myeloid populations including MDSCs, DCs and Granulocytes/Neutrophils or lymphocytic response. That may be a strength of this review, however, less detail is provided compared to TAMs. The discussion of so many immune cell types in a short review leave many gaps in literature. Would choose to focus on either myeloid or non-myeloid cell types to improve the review.  

The review offers that macrophages and microglia are interchangeable terms and this is not necessarily correct and would be confusing to readers. Separate microglia and macrophages and describe more recent evidence of differences.

Review is missing some myeloid targets such as don’t eat me signals including CD47/SIRPa.

The review only offers discussion about glioma and immune response in the context of no treatment. Post treatment especially post radiation immune response is also critical in glioma and would be interesting to reviewer. If focused on a subset of immune populations a post treatment discussion could be included.

The review uses glioma generally, however, there is a large difference in immune response of low grade and high grade glioma. Would choose LGG or HGG to focus the review better and be more clear which is being discussed.

Lymphocytes are not an abundant cell type in the immune microenvironment of gliomas, especially HGG. Therefore, would only use the discussion of these cells types in context of the myeloid populations focused on.

The context of immunotherapies is confusing. Would remove or include as part of each cell type discussion. “Treatment options targeting MDSCs” for example. This would provide evidence that the immune response discussed is clinically relevant.

Minor Comments

Title is too broad.

Myeloid cells:

-        Figure 1 is good except missing some signals i.e. “don’t eat me”. May be better to show direct vs indirect if it’s too much in one figure.  

TAMs

-          First sentence of 2.1.1 is odd and confusing

-          Line 79 – no comma after Chen et al.

Granulocytes

-          Are neutrophils the only relevant granulocyte? – others aren’t mentioned. If yes, then don’t use the term granulocyte or make that explicit.

Dendritic cells

-          DCs respond to type 1 interferon, alpha and beta, but this is not discussed in the review.

-          Discussion of DC function in other tumor types like glioma Ref 66 but not clear that it’s another tumor.

-          Be more clear what has been found in glioma vs other tumors then comment on  what needs to be explored compared to other tumor types if it hasn’t already.

MDSCs

-          Line 258 – “NO has can upregulate Notch…” à get rid of has

-          Lines 267-268 – sentence doesn’t make sense

T cells

-          Discuss PD-L1 / PD-1 interaction in regards to T cell inhibition, but don’t talk about PD-L1/PD-1 in don’t eat me context with myeloid

-          Same with B2M on MHC-I – discuss it in relation to infiltrating lymphocytes but not as a myeloid don’t-eat-me signal

Cytokine and molecular therapies - Lines 380-383 – how does both inhibition and activation of PP2a have the same effect on STING and type 1 IFN response?

Immune checkpoint inhibitors – missing checkpoints that target phagocytosis or “don’t eat me” signals

CAR T-cell therapy – largely used in pediatric glioma but context of this is not included either here or in introduction. Would suggest either choosing to include pediatrics or not but don't combine pediatric and adult glioma without mentioning that they are not the same. 

Conclusions and perspectives:  would need to be re-written depending on the above decisions for focus of the review.

Other paper to consider including:

-        The crosstalk between tumor cells and the immune microenvironment in breast cancer: implications for immunotherapy; https://www.ncbi.nlm.nih.gov/pmc/articles/PMC7991834/

Would strongly consider using papers in BioRx to improve manuscript - https://www.biorxiv.org/content/10.1101/2023.10.24.563466v1 is a fantastic new paper focused on myeloid cell types

Comments on the Quality of English Language

No concerns

Author Response

The authors prepared an interesting review article, however, the first major concern is that the review is both too broad and in some places too narrow to be a review on a specific topic. Would advise the authors to pick a focus of interest to narrow in on to provide a more comprehensive and appropriate review.

Of note, there are multiple review articles on TAMs and glioma/glioblastoma. There are less review articles focused on other myeloid populations including MDSCs, DCs and Granulocytes/Neutrophils or lymphocytic response. That may be a strength of this review, however, less detail is provided compared to TAMs. The discussion of so many immune cell types in a short review leave many gaps in literature. Would choose to focus on either myeloid or non-myeloid cell types to improve the review.  

We thank the Reviewer for the thoughtful comments and are delighted the Reviewer found our review interesting. In this review, we aimed to capture the main well-described and validated immune-glioma cross-talk pathways that lead to an immunosuppressive TME and glioma progression. In doing so, we believe it is important to highlight both the myeloid and lymphoid immune cell populations since immune cell-specific interactions with glioma cells not only affect the specific immune cell type but have global implications on the immune TME. We agree that the majority of glioma immunology research to date has focused on TAMs, and thus, this section in our review is the most detailed, as the Reviewer noted. Furthermore, as the Reviewer comments below, the lymphoid cell population in the TME is marginal compared to myeloid cells which may be another reason they are less studied currently. Our assessment of the current strong evidence for the role of non-TAM immune cell types would conclude that more in-depth research focused on these cells is needed prior to centering a review on just these cell-types. As such, we would prefer to keep the organization and focus of our review as is. We have now included the following sentence in the Conclusions and Perspectives section to further emphasize this point (Pg. 13, lines 525-528): “While significant progress has been made in characterizing the components of the TME, the majority of current studies have focused on TAM population of immune cells, given their abundance in the TME, and further in-depth mechanistic studies on the role of other immune cell types in glioma biology is needed.”

The review offers that macrophages and microglia are interchangeable terms and this is not necessarily correct and would be confusing to readers. Separate microglia and macrophages and describe more recent evidence of differences.

We thank the Reviewer for their insightful comment and agree that recent work has begun to highlight differences in microglia (MG) and bone-marrow-derived macrophages (BMDMs). It is indeed an important point, and we now included a discussion on the distinct cellular origins of MG and BMDM as well as the emerging evidence that they differentially contribute to the immunosuppressive environment of glioma:

Pg. 2, lines 55-62: “Macrophages in the brain consist of both bone-marrow-derived macrophages (BMDMs) and resident macrophages of the central nervous system (CNS) termed microglia. Studies have revealed that microglia originate from yolk-sac progenitors during embryogenesis and have distinct transcriptional states depending on their topological distribution in the human brain [9-11]. While the exact differences in microglia and BMDMs remain to be elucidated, these macrophage types modulate immune responses through pathogen phagocytosis and antigen presentation and also function in wound healing and tissue repair.”

Pg. 2, lines 74-88: “Recent studies have begun to try to distinguish between microglia and BMDMs in glioma formation and progression. Lineage-tracing experiments in mouse glioblastoma models have suggested that 85% of TAMs are BMDMs, which are predominantly localized in the perivascular areas of the tumor, while microglia are peri-tumoral [19]; however, other studies have suggested that the majority of TAMs are intrinsic microglia [20]. In-terestingly, studies comparing the type of TAM (i.e. microglia vs. BMDM) in IDH-mutant and wild-type tumors have suggested that IDH-mutant TAMs were composed primarily of mi-croglia, while IDH-wild-type tumors had more BMDM infiltration [12,21]. In astrocytomas, increased infiltration of TAMs derived from BMDM was associated with higher tumor grade and reduced survival [22]. More in-depth studies are needed to accurately distinguish microglia from BMDMs and further elucidate their distinct roles within specific types of gliomas. Given the current lack of clear evidence on the differences of roles between microglia and BMDMs in glioma, our connotation of TAMs in this review denotes mac-rophages without distinction between these two myeloid cell populations.”

Review is missing some myeloid targets such as don’t eat me signals including CD47/SIRPa.

We thank the Reviewer for this suggestion and have now included the CD47 axis as an important myeloid checkpoint:

Pg. 3, lines 125-129: “In addition to reprogramming of TAMs, glioma cells can upregulate the antiphagocytic “don’t eat me” surface protein CD47 which binds to its receptor SIRP on TAMs and in-hibits their phagocytic activity [38,39]. Blocking this myeloid checkpoint CD47-SIRP axis using anti-CD47 antibody therapy was shown to limit tumor growth in both adult and pe-diatric gliomas [40,41] and also promoted M1 activation of TAMs [42].”

The review only offers discussion about glioma and immune response in the context of no treatment. Post treatment especially post radiation immune response is also critical in glioma and would be interesting to reviewer. If focused on a subset of immune populations a post treatment discussion could be included.

While we agree that changes in the immune TME associated with treatment and recurrence is an interesting topic, a detailed discussion on this issue is beyond the scope of this review. We note that we have now included in the Conclusions and Perspectives section a brief mention for this important future direction (Pg. 13, lines 515-519): “In addition, recent studies have begun to examine changes in the immune TME associated with standard of care GBM therapies temozolomide and radiation [190]. Understanding how these treatments affect the TME and how gliomas may acquire resistance to these therapies through immune cell adaptations may help develop more efficacious combi-nation treatments.”

The review uses glioma generally, however, there is a large difference in immune response of low grade and high grade glioma. Would choose LGG or HGG to focus the review better and be more clear which is being discussed.

We agree there are immune cell type and response differences between LGG and HGG and have now highlighted examples in the manuscript as well as indicated when appropriate which glioma type was studied in articles cited.

Pg. 2, lines 79-82: “Interestingly, studies comparing the type of TAM (i.e. microglia vs. BMDM) in IDH-mutant and wild-type tumors have suggested that IDH-mutant TAMs were composed primarily of microglia, while IDH-wild-type tumors had more BMDM infiltration [12,21].”

Pg. 8, lines 321-325: “PD-L1 expression was noted to be lower in IDH-mutant, low-grade gliomas (LGG) [119], which also had reduced expression of T-cell lymphocyte associated genes compared to IDH-wild-type, high-grade gliomas (HGG) [122]. Further investigation is needed to elucidate the mechanisms contributing to these differences in LGG versus HGG check-point molecule expression.”

Lymphocytes are not an abundant cell type in the immune microenvironment of gliomas, especially HGG. Therefore, would only use the discussion of these cells types in context of the myeloid populations focused on.

We agree with the Reviewer that the immune TME is lymphocyte-deficient; nonetheless, there are limited studies that have investigated glioma-driven mechanisms that underlie the dysfunction of T cells in glioma progression, and we believe they should be highlighted in the review (in addition to TAM/MDSC-mediated repression of T cells which we also discuss). As such, we would prefer to keep the lymphoid section as is and now further emphasize in the Conclusions and Perspective the need for additional in depth studies focusing on these immune cell types (see comment #1).

The context of immunotherapies is confusing. Would remove or include as part of each cell type discussion. “Treatment options targeting MDSCs” for example. This would provide evidence that the immune response discussed is clinically relevant.

We appreciate the Reviewer’s comment; however, we believe keeping the immunotherapy discussion as it currently is in a separation section is preferable to the reader given that not each cell type has well-studied immunotherapies and some immunotherapies may have downstream effects on multiple cell types (e.g. oncolytic viruses promoting a pro-inflammatory TME rather than specifically targeting one immune cell type). 

Minor Comments

Title is too broad.

We believe the title encapsulates our scoping review of the crosstalk pathways between glioma and immune cells that promote tumor progression. As such, we would like to keep the title as.

Myeloid cells:

      Figure 1 is good except missing some signals i.e. “don’t eat me”. May be better to show direct vs indirect if it’s too much in one figure.  

      We have now included the CD47/SIRPa interaction in Figure 1.

TAMs

      First sentence of 2.1.1 is odd and confusing

We have now removed this sentence.

      Line 79 – no comma after Chen et al.

We have now removed this comma.

Granulocytes

      Are neutrophils the only relevant granulocyte? – others aren’t mentioned. If yes, then don’t use the term granulocyte or make that explicit.

Since other granulocytes are not well studied, we have now changed the sub-title to state “Neutrophils”.

Dendritic cells

      DCs respond to type 1 interferon, alpha and beta, but this is not discussed in the review.

We have now noted DC activation to type 1 interferon (pg 6, line 227-230)

      Discussion of DC function in other tumor types like glioma Ref 66 but not clear that it’s another tumor.

We have modified the text to indicate that these studies were done in metastatic melanoma brain tumors in addition to primary brain tumors.

      Be more clear what has been found in glioma vs other tumors then comment on  what needs to be explored compared to other tumor types if it hasn’t already.

We have modified the text to clarify studies not performed in gliomas. 

MDSCs

      Line 258 – “NO has can upregulate Notch…” à get rid of has

We have modified the text as suggested.

Lines 267-268 – sentence doesn’t make sense

We have modified the text as suggested.

T cells

Discuss PD-L1 / PD-1 interaction in regards to T cell inhibition, but don’t talk about PD-L1/PD-1 in don’t eat me context with myeloid

We note that PD-L1/PD-1 interaction leads to inhibition of T cell differentiation and activation.

      Same with B2M on MHC-I – discuss it in relation to infiltrating lymphocytes but not as a myeloid don’t-eat-me signal

We have noted that loss of B2M expression is a mechanism tumors use to reduce MHC-I antigen expression.

Cytokine and molecular therapies - Lines 380-383 – how does both inhibition and activation of PP2a have the same effect on STING and type 1 IFN response?

The sentence only notes inhibition of PP2a leading to STING activation and induction of type I IFN response.

Immune checkpoint inhibitors – missing checkpoints that target phagocytosis or “don’t eat me” signals

As suggested, we now also note the anti-CD47 therapy in this section (in addition to the TAM section).

CAR T-cell therapy – largely used in pediatric glioma but context of this is not included either here or in introduction. Would suggest either choosing to include pediatrics or not but don't combine pediatric and adult glioma without mentioning that they are not the same. 

We have found several CAR T therapy studies trialed for GBM in addition to diffuse midline gliomas and thus would prefer to include these studies highlighting CAR T in both adult and pediatric populations.

Conclusions and perspectives:  would need to be re-written depending on the above decisions for focus of the review.

We have edited the Conclusions and Perspectives section as indicated in the above comments.

Other paper to consider including:

The crosstalk between tumor cells and the immune microenvironment in breast cancer: Implications for immunotherapy; https://www.ncbi.nlm.nih.gov/pmc/articles/PMC7991834/

Would strongly consider using papers in BioRx to improve manuscript - https://www.biorxiv.org/content/10.1101/2023.10.24.563466v1 is a fantastic new paper focused on myeloid cell types

We thank the Reviewer for pointing us to this bioRx paper which is indeed very interesting and have now included it in our Conclusions and Perspective section.

Reviewer 2 Report

Comments and Suggestions for Authors

As the authors claim, they aim to provide an updated view of the mechanisms underlying glioma-immune cell interactions and novel therapies being developed to combat glioma immune evasion. Some points should be noted as below.

1) Perhaps we can discuss “cancer, TME and immune cell interactions from a new perspective. A new paper proposes that cancer is an ecological disease that is multidimensional spatiotemporal , and cancer cells and TME including immune cells is a complex pathological ecosystem (Theranostics 2023; 13(5):1607-1631. https://pubmed.ncbi.nlm.nih.gov/37056571/). This paper is suggested to be reviewed and such view should be updated.

2) If we consider that “: cancer is a disease with multidimensional spatiotemporal "unity of ecology and evolution", it is obviously not enough to explain the occurrence and development of cancer only from the molecular point of view. How about mesoscale between tumor cells and immune cells (independent of molecular mechanisms)?

3) From above, Figure 1 “The bidirectional, cellular interactions between myeloid and glioma cells” from molecular level is not sufficient. How about the relationships between  myeloid and glioma cells, or between myeloid cell themself (e.g. competition, predation, parasitism or mutualism) ?

4) How about underlying glioma-immune cell interactions in vivo model? 

Author Response

As the authors claim, they aim to provide an updated view of the mechanisms underlying glioma-immune cell interactions and novel therapies being developed to combat glioma immune evasion. Some points should be noted as below.

1) Perhaps we can discuss “cancer”, “TME” and immune cell interactions from a new perspective. A new paper proposes that cancer is an ecological disease that is multidimensional spatiotemporal , and cancer cells and TME including immune cells is a complex pathological ecosystem (Theranostics 2023; 13(5):1607-1631. https://pubmed.ncbi.nlm.nih.gov/37056571/). This paper is suggested to be reviewed and such view should be updated.

We appreciate the Reviewer’s interesting perspective and the cited paper. We agree that this is an important consideration for tumor biology as cancer is not a static disease but rather continuously adapts to environmental changes as we had noted in our Conclusions and Perspectives section (Pg. 13, lines 511-517): “While the specific mechanisms underlying the evolution of glioma tumors remain to be solved, spatial transcriptomic and proteomic evidence has shown that glioma cells are dynamically reorganized into distinct regions that display unique molecular profiles or lineage states (e.g., proneural, mesenchymal, classical.) and are constantly adapting to evolving stresses from the surrounding TME [58,190]. Thus, characterization of gliomas in patients cannot be a singular event and requires multiple regions and time points to be analyzed.”

We have further added additional sentences at the beginning of the Conclusions and Perspectives section to include this paper and give a broader overview of this perspective and how it incorporates into our Review (Pg. 13, lines 495-500): “Cancer is no longer simply regarded as a genetic disease that leads to the formation of tumorigenic cells. Instead, we have come to appreciate the “ecological” aspect of this disease where tumor development is intimately linked to reciprocal interactions with non-cancerous cells in their surrounding microenvironment [187]. Spatiotemporal in-teractions between these cells leads to adaptations and evolution of cancer as it progresses from initiation to immune evasion and invasion.”

2) If we consider that “: cancer is a disease with multidimensional spatiotemporal "unity of ecology and evolution", it is obviously not enough to explain the occurrence and development of cancer only from the molecular point of view. How about mesoscale between tumor cells and immune cells (independent of molecular mechanisms)?

This is indeed an interesting perspective, however, this is beyond the scope of our review which focuses on the molecular pathways underlying immune cell and glioma cross-talk. There are very recent studies that have investigated the spatiotemporal heterogeneity of gliomas and associated differences in the TME which we have cited in the Conclusions and Perspectives (as indicated in the comment above). 

3) From above, Figure 1 “The bidirectional, cellular interactions between myeloid and glioma cells” from molecular level is not sufficient. How about the relationships between  myeloid and glioma cells, or between myeloid cell themself (e.g. competition, predation, parasitism or mutualism) ?

We appreciate the Reviewer’s interesting question. We do note in our manuscript enrichment of certain cell populations (e.g. TAMs, MDSCs, and Tregs) are associated with higher tumor grade, invasiveness and depletion of other immune cell types (e.g. Tregs can reduce cytotoxic T cell infiltration), and we highlight the molecular interactions between glioma and these cell types that are tumor-supportive and immune-cell supportive (i.e. mutualistic relationship). However, because these interactions are dynamic and studies have not characterized these interactions using “ecological” terms, we would prefer to describe these interactions as tumor-supportive or anti-tumor responses.

4) How about underlying glioma-immune cell interactions in vivo model? 

Our review includes in vivo data (both in human and in mouse glioma models) of glioma-immune cell interactions.

Reviewer 3 Report

Comments and Suggestions for Authors

Firstly, I would like to congratulate the authors on their well-written manuscript. The article titled “Glioma-immune cell crosstalk in tumor progression “ is meticulously crafted, presenting an updated and in-depth analysis of the mechanisms underlying glioma and immune cell interactions within the Tumor microenvironment (TME). The authors have done an exceptional job in synthesizing current knowledge on both pro- and anti-tumor activities of myeloid and lymphoid populations in the glioma TME. Their exploration of the complex crosstalk between cancer and immune cells, particularly in how glioma evades the immune system and co-opts these cells for tumor growth, is both insightful and significant. A standout feature of this review is it’s comprehensive coverage of current and experimental therapeutic strategies aimed at countering the immunosuppressive activity of the glioma TME. The authors present a compelling rationale that a deeper understanding of the intricate interplay between tumor and immune cells could pave the way for novel combination treatments.

Furthermore, the article is exceptionally well-referenced, with the authors having meticulously included the most up-to-date and relevant references. In summary, the review article presents a comprehensive, well-structured, and insightful analysis of the dynamic interactions within the glioma TME. The depth of research, combined with the clarity of presentation, makes this article a noteworthy contribution to the field. I highly recommend this article for publication.

Author Response

We thank the Reviewer for these supportive comments.

Round 2

Reviewer 2 Report

Comments and Suggestions for Authors

Good, no other questions.